# Lifestyle risk factors and metabolic markers of cardiovascular diseases in Bangladeshi rural-to-urban male migrants compared with their non-migrant siblings: A sibling-pair comparative study

Shirin Jahan Mumu [1,2]*, A. K. M. Fazlur Rahman[2], Paul P. Fahey[1], Liaquat Ali[3], Dafna Merom[1]

1 School of Health Sciences, Western Sydney University, Sydney, Australia, 2 Dept of Epidemiology, Bangladesh University of Health Sciences (BUHS), Dhaka, Bangladesh, 3 Pothikrit Institute for Health Studies, Dhaka, Bangladesh

* shirinmumu@yahoo.com

**Data Availability Statement:** All files are available from the OPENICPSR database https://www.

## Abstract

### Background

The increasing prevalence of cardiovascular diseases (CVDs) in developing countries like Bangladesh has been linked to progressive urbanisation. Comparisons of rural and urban populations often find a higher prevalence of CVD risk factors in the urban population, but rural-to-urban migrants might have different CVD risk profiles than either rural or urban residents. This study aimed to describe differences in CVD risk factors between migrants and non-migrants siblings and to determine whether acculturation factors were associated with CVD risk factors among migrants.

### Methods

Using a sibling-pair comparative study, 164 male migrant who migrated from Pirganj rural areas to Dhaka City and their rural siblings (total N = 328) were assessed by interview, anthropometric measurement, blood pressure and blood samples. Comparisons were made using linear or logistic mixed effects models.

### Findings

Physical inactivity, inadequate intake of fruit and vegetables and possible existence of a mental health disorder had 3.3 (1.73; 6.16), 4.3 (2.32; 7.92) and 2.9 (1.37; 6.27) times higher odds among migrants than their rural siblings, respectively. Migrants watched television on average 20 minutes (95% CI 6.17–35.08 min/day) more per day than the rural sibling group whereas PUFA intake, fruit and vegetable and fish intake of the migrants were -5.3 gm/day (-6.91; -3.70), -21.6 serving/week (-28.20; -15.09), -14.1 serving/week (-18.32; -9.87), respectively, lower than that of the rural siblings. No significant difference was observed for

openicpsr.org/openicpsr/workspace?goToPath=/
openicpsr/174241&goToLevel=project#.

**Funding:** AKMFR, SJM- Education Ministry of
Bangladesh SJM- Western Sydney University HDR
RTS Fund The funders had no role in study design,
data collection and analysis, decision to publish, or
preparation of the manuscript.

**Competing interests:** The authors have declared
that no competing interests exist.

other variables. After adjusting, the risk of physical inactivity, inadequate fruit and vegetable
intake, a mental health disorder and low HDL were significantly higher in migrants than in
rural siblings and tended to be higher for each increasing tertile of urban life exposure.

## Conclusion

The findings suggest that migration from rural-to-urban environment increases CVD risk
which exacerbate with time spent in urban area due to acculturation. This study gives new
insights into the increased CVD risk related with migration and urbanization in Bangladesh.

## Introduction

Cardiovascular disease (CVD) is a major health problem across the world accounting for 32%
of all deaths in 2019 and over 75% of CVD deaths occur in low and middle income countries
[1]. Demographic transition (from declining fertility and increased life expectancy) accompa-
nied by urbanisation are driving transformations in lifestyle such as nutrition habits and physi-
cal activity (PA) which change the risk of CVD [2–4].

It has been long observed that when a person migrates to the city from rural area s/he faces
numerous changes, not only in the socio-cultural environment but also in lifestyle such as diet
and physical activity [5, 6]. Various studies showed that rural-to-urban migrants had higher
prevalence of CVD risk factors [7–11]. In India, rural-to-urban migrants reported higher
energy intake, such as macronutrient (carbohydrate, protein and fat) and sugar intake, com-
pared with non-migrant and rural counterparts [12]. Studies from Guatemala [13] and China
[14] showed that migrants consumed more fat and cholesterol than non-migrant rural coun-
terparts and the Chinese study also found lower dietary fibre intake in migrants. Similarly,
studies demonstrate that urban people and migrants from rural-to-urban areas had lower lev-
els of physical activity than rural people [10, 15].

The global burden of disease (GBD) study demonstrated that most developing countries
have been experiencing an increasing prevalence of unhealthy behaviours accompanied by
changes in underlying metabolic and physiological CVD risk factors [16]. Both these types of
risk factors, often coexist in the same individual, and work synergistically to increase the indi-
vidual's total risk of developing acute vascular events such as heart attack and stroke [17].

Bangladesh is experiencing rapid urbanisation, mainly through rural-to-urban migration
[18]. The population in Dhaka is growing by an estimated 4.2% per year, one of the highest
rates amongst Asian cities [19]. This is likely to have profound implications for its population
health profile. The WHO STEPs surveys of Bangladesh (2006, 2010, 2013 and 2018) provide
evidence that both behavioural risk factors (e.g. low intake of fruit/vegetables and physical
inactivity) and metabolic risk factors (e.g. overweight, hypertension and documented diabetes)
have been increasing [20–23].

Despite this rapid urbanisation, there is currently no study that has focused on changes in
CVD risk factors with internal migration in Bangladesh. Few nationwide cross-sectional sur-
veys have compared dietary intake and physical inactivity between rural and urban areas [21,
24, 25]. However, results are descriptive in nature and do not adjust for possible socio-eco-
nomic moderators, such as income, education or occupation, that may further explain the
rural-urban differences. Moreover, the cross sectionals rural-urban comparison cannot disen-
tangle the cultural, genetic and lifestyle background of the migrants from the host population
which may or may not be similar [8].

Migration is further complicated as it may be influenced by 'selection of migrants' according to higher or lower risk of health and disease [8, 26]. In addition, as most of the migration was due to economic reasons, it is assumed that those with better health or lifestyle and socio-economic status are more likely to be able to afford migration. In this scenario, the lifestyles and health profile of high socio-economic status (SES) individuals have already been determined in the pre-migration period [26].

A sibling-pair study design can better address many of the limitations of the cross-sectional comparisons, and also avoid the cost associated with a prospective cohort study; following up migrants for many years to detect changes. A sibling-pair study limits the demographic, cultural or behavioural differences between rural and migrant groups as it assumed the sibling pairs share many health determinants from their past, including environmental and genetic factors. Hence the differences observed at present between the migrant and the rural sibling represent divergence from the shared lifestyle and environment change.

Quantifying the health risk associated with rural to urban migration was required to assist in guiding and evaluating health interventions. A sibling-pair comparative study was conducted to assess differences in CVD risk factors between male migrants and non-migrants siblings in Bangladesh and to determine whether acculturation factors were associated with CVD risk factors among migrants.

## Methods

### Ethics approval

This study was approved by the Western Sydney University Human Research Ethics Committee (HREC # H11056) and Bangladesh University of Health Science Ethical Review Committee. Written consent was obtained from all participants before data collection.

### Study design

A sibling-pair comparative study design [10] was used to compare rural-to-urban migrants to their rural siblings. The advantage of this design is primarily the controls for the genetic predisposition of the CVD and the environmental influences within the family and the surrounding environment. It assumes that the lifestyles prior to migration are similar between the two siblings but diverge once one sibling migrates to an urban area and the other remains residing in the original rural area.

### Place of study

This study was conducted in the capital city, Dhaka, and the Pirganj subdistrict/upazila of Thakurgaon district of Bangladesh, which were selected conveniently. Dhaka is the capital and one of the largest cities in Bangladesh. The population of Dhaka Metropolitan is around 8.9 million [27].

The Thakurgaon district is situated in the northern part of Bangladesh. This district has five subdistricts: Baliadangi, Haripur, Pirgonj, Ranisankail and Thakurgaon Sadar. This study was conducted in the Pirganj subdistrict which has an area of 354 square kilometres and consists of 10 unions with the total number of residents reaching 2,43,535 [28]. It is far from Dhaka (390 km) and urbanization was only 11.4%, hence it maintains its rural characteristics.

### Study population

Two groups were selected; **Rural-to-urban migrants**, who migrated from Pirganj to Dhaka City and had been residing there permanently for at least one year and were first generation

migrants and had a non-migrant sibling of the same gender; and **Rural**, participants who have always lived in a rural environment or in Pirganj. Those eligible to participate were aged 18–60 years, either gender and consenting to participate.

Those with an intellectual disability, or those with any chronic medical condition which required dietary and/or physical restriction were excluded. Temporary migrants, such as seasonal workers or visitors and migration for medical reasons, were excluded from the study.

## Sample size

As dichotomous variables generally require the greatest sample sizes, the required sample size was first calculated for the dichotomous variable, physically active or sedentary lifestyle. As the data are paired (siblings), analysis was based on McNemar's Chi-square test. It was anticipated that there would be a strong tendency towards more sedentary behaviour in urban compared to rural areas. Hence, calculation of sample size was set to detect a minimum odds ratio of 2.0. It was anticipated up to 50% of pairs would have both siblings displaying the same behaviour (active or sedentary) leaving 50% of pairs discordant. In total, 144 pairs were required for a two sided McNemar's test to have at least 80% power to detect an odds ratio of 2.0 or larger at the 5% significance level. After allowing for a 15% non-response rate, the sample size was increased to 169 pairs resulting in total 346 participants (169x2 = 338). This calculation was done using the sample size calculator G*Power v3.1.9 [29].

For analyses of mean difference between siblings on continuous variables, assuming a 2-sided paired t-test and either 0.2, 0.5 or 0.8 correlation between siblings, the target sample size of 144 pairs provided at least 80% power to detect at least a 0.24 standard deviation difference between groups. Thus, according to Cohen's criteria [30], the study was powered to detect a small difference between groups.

## Phases of the study

The study was conducted in two phases;

**The first phase was the rural Household (HH) survey.** This survey was conducted in the Pirganj subdistrict to identify households where at least one person had migrated from the villages. This phase was undertaken in 26 randomly selected mouzas (i.e., villages) of at least 300 households to represent the subdistrict. Every household in the village was interviewed and detailed household information was taken from the family head. If there was any migrant in the household, detailed migration related information such as duration, reason, place etc. was obtained.

**Second phase was the migrant-sibling study.** In this stage all identified households were selected for data collection on CVD risk factors if any family member or relative was residing in urban areas of Dhaka. Contact details of migrants were obtained from their family members and a list of potential migrant participants was created. Each migrant to Dhaka on the list was contacted to confirm their eligibility and obtained consent from both siblings before enrolment. (See S1 File for the sampling technique in detail).

## Data collection tools

A semi-structured, pre-tested, interviewer administered questionnaire was used to collect information. The survey instrument included a pre validated food frequency questionnaire (FFQ) [31] and the global physical activity questionnaire (GPAQ) Version 2 translated into Bangla (Bengali) and validated on the target population [32]. The FFQ asks about the frequency of consumption over the last three months and portion size of commonly consumed food items. Detailed nutrient values for Bangladeshi foods [33] were used to calculate calorie,

total fat and Polyunsaturated fatty acids (PUFA) intake. Additional questions included demographic and socio-economic variables, smoking status, alcohol and smokeless tobacco consumption. As estimation of income in a rural area is complex, multiple questions were asked including monthly income and expenditure, yearly income and sources such as farming, business, rent, remittance etc, and ownership of home, land and assets. The Kessler Psychological Distress Scale (K10) [34] was used to assess the presence of common mental disorders among participants. Migration related information was obtained including years living in urban area, and degree of acculturation. Acculturation was measured by use of language, current dietary practice compared to before migration and perception of change in physical activity from before to after migration and length of urban stay.

A checklist was also included to record biochemical tests (fasting blood sugar and lipid profile), clinical variables (SBP, DBP) and anthropometric variables (height, weight, waist circumference, hip circumference, skinfold thickness: bicep, tricep, subscapular and suprailiac). Interview, anthropometry and blood pressure measurement were conducted by trained research assistants.

## Anthropometry and blood pressure measurement technique

Weight was measured using an electronic digital LCD weighing machine and height was measured using a height measuring tape. Waist circumference was taken by placing a measuring tape horizontally midway between the lowest rib margin and the iliac crest in the mid axillary line to measure waist circumference to nearest centimetre. Hip circumference was measured by measuring tape at the widest part of the buttocks or hip. Skinfold thickness (SFTs) was measured using Harpenden Skinfold Calipers (BATY International Limited, United Kingdom) to the closest 0.2 mm on to the right side of the body. Measurements were taken at the triceps, biceps, subscapular and suprailiac sites.

Blood pressure was measured in a sitting position, with cuff at the level of the heart using a sphygmomanometer. After 5 minutes of rest a second reading was taken. If the difference between the two readings was more than 5 mm of Hg for SBP and DBP, a third reading was taken. The mean of the three readings was then considered as the final blood pressure of the participant for data analysis.

## Collection of blood samples for biochemical measures

For urban participants the blood sample was collected on the day of interview at the Bangladesh University of Health Sciences (BUHS) hospital. Rural participants were invited to attend a blood collection camp for blood sample collection. The invitation listed the name of the participant, their age, study identification number, name of their village, and the scheduled time, date and location for their blood sample collection. They also received verbal instructions. Five blood collection camps were conducted in Pirganj subdistrict for rural participants. A transport allowance was reimbursed to all rural and urban participants for attending at the BUHS hospital and camp.

The venous blood was drawn by a trained phlebotomist after at least eight hours of overnight fasting. Five millilitres of blood was taken out for fasting glucose and lipid profile. After that, participants were given 75 grams of glucose in 250 ml of water to drink. Another three millilitres of blood was collected after two hours for 2-h post-oral glucose tolerance test (OGTT). It was ensured that the participant was not doing any vigorous physical activity or taking any food in this time period.

After 30 minutes, blood samples were centrifuged for 10 minutes at 3000 rpm to obtain serum. Tests of all samples were performed at the BUHS laboratory. Samples from rural

participants were transferred with Ethylenediaminetetraacetic acid (EDTA) to the core laboratory at BUHS in a box containing dry ice to maintain a suitable temperature. All samples were preserved in a freezer (-70 C) until the laboratory assays were carried out. Serum glucose was measured by the glucose-oxidase method and the serum lipid profile [total cholesterol (TC), triglyceride (TG), and high density lipoprotein cholesterol (HDL-c)] was measured by the enzymatic-colorimetric method using a conventional automated analyser (Dimension® clinical chemistry system, Siemens Healthcare Diagnostics Inc. USA). The LDL-Cholesterol level in serum was calculated using the following formula [35];

$$LDL - cholesterol = Total\ cholesterol - [1/5(Triglycerides) + HDL\ cholesterol].$$

## Data analysis

Histograms and descriptive statistics were used to review the distributions of continuous variables for departures from symmetry. Frequency counts and percentages were used to check categorical variable for categories with insufficient observations.

Socioeconomic classifications were made according to the 2006 per capita Gross National Income (GNI) and according to World Bank (WB) calculations [36]. Physical activity level (high, moderate, low) was categorised according to the Global Physical Activity Questionnaire (GPAQ) scoring protocol [37]. Inadequate fruits and vegetable intake was defined as <5 serving/day [38]. Smoker or smokeless tobacco user was defined as smoked or used smokeless tobacco currently or within the last 3 month or occasionally. Probable case of mental health disorder was defined if Kessler 10 score was ≥20 [34]. Asian BMI criteria were used to categorise and define underweight (< 18·5 kg/m$^2$), normal (18.5–23.0 kg/m$^2$), overweight (23–27.5kg/m$^2$), and obese (> 27.5 kg/m$^2$) [39]. Abdominal obesity was diagnosed when waist circumference was >90 cm in men using the WHO definition [40]. Dyslipidaemia was defined as high total serum cholesterol (≥200 mg/dl), high triglycerides (≥150 mg/dl), high LDL cholesterol (≥130 mg/dl) or low HDL cholesterol (<40 mg/dl in men). Hypertension was defined as a systolic blood pressure ≥ 140 mmHg or a diastolic blood pressure ≥ 90 mmHg or current treatment with antihypertensive medication (2.5% of the total participants) [41]. Diabetes was defined if FBG ≥7.0 mmol/L or OGTT ≥11.1 mmol/L or self-reported diabetes medication use (5% of the total participants) [42].

Paired t-tests, Wilcoxon signed-rank tests and McNemar Chi-square tests were performed to compare between group differences in paired continuous normal, non-normal and categorical outcome variables, respectively. For multivariate models, a random effect of pair was included which allowed the variation between two siblings (within-pair variation) to differ from the variation between non-siblings (between-pair variation). Comparisons were made between rural-to-urban migrants and rural siblings using linear mixed effects models for continuous outcomes and logistic mixed effect models for the binary outcomes. The association between length of urban residence, categorised into tertiles, and CVD risk factors were assessed by logistic regression model controlling for potential confounding factors. In this model, rural siblings are taken as reference group as zero urban life years.

All p values presented were two tailed. The statistical tests were considered significant at a level of 5% (0.05).

## Results

### Response rate

Twenty-six villages of 10 unions of Pirganj subdistrict were randomly selected and a total of 13,736 household were visited to identify rural-to-urban migrants. Compared to the

Bangladesh Census 2011, the coverage of this household survey was 98.7%. A total of 452 migrants were identified who migrated to Dhaka city and among them 429 migrants were contacted.

After completing the eligibility criteria, the final sample size was 176 pairs. Of these, only 7% (12/176) were female. Hence further analysis was conducted on men only (164 pairs). Overall, the response rate at completion of the study was 63%. The selection of study participants is described in Fig 1.

## Study population characteristics by migration status

The demographic characteristics of the migrants and their siblings are summarised and compared in Table 1. Compared to migrants, the mean±SD age was higher for rural men (31.9±7.5

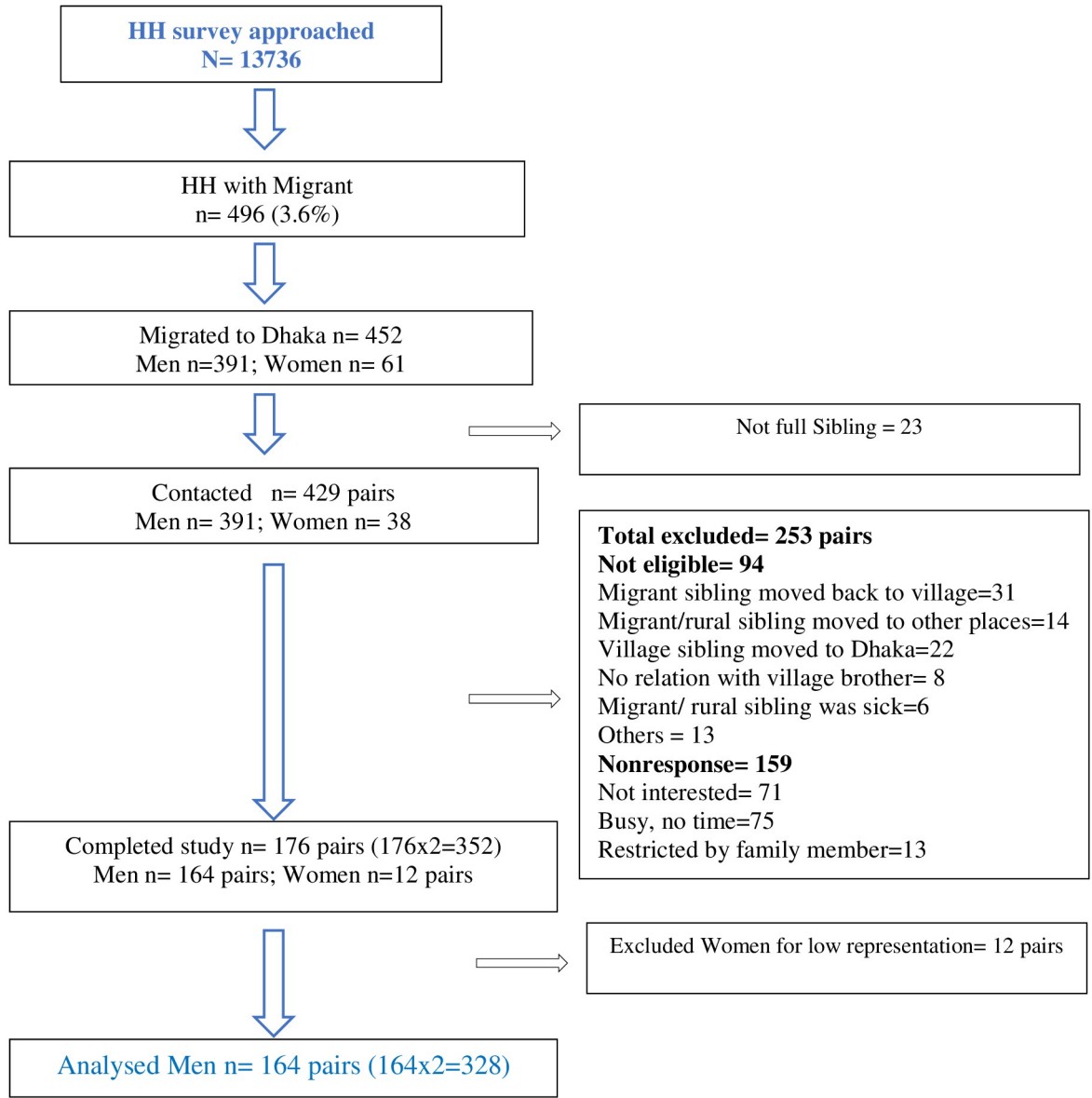

**Fig 1. Selection of study participants flowchart.**

**Table 1. Study population characteristics by migration status.**

| Study Population Characteristics | Rural-to-Urban Migrants | Rural Siblings | p |
|---|---|---|---|
| Age (y), mean (±SD) | 31.87 (±7.54) | 33.35 (±9.33) | 0.02* |
| Muslim Religion, n (%) | 147 (89.6) | 147 (89.6) | 1.00† |
| Marital status, n (%) | | | 0.76† |
| Currently married | 119 (72.6) | 116 (70.7) | |
| Never married | 45 (27.4) | 48 (29.3) | |
| Education level, n (%) | | | <0.001† |
| Nil to Primary level | 24 (14.6) | 52 (31.7) | |
| High school level | 111 (67.7) | 88 (53.7) | |
| University level | 29 (17.7) | 24 (14.6) | |
| Occupation, n (%) | | | 0.02† |
| Manual (farmer, day labour, factory workers etc.) | 87 (53.0) | 108 (65.9) | |
| Others (professional, teacher, clerk, service etc.) | 77 (47.0) | 56 (34.1) | |
| House type | | | <0.001† |
| Mud | 2 (1.2) | 58 (35.4) | |
| Tin shed | 71 (43.3) | 81 (49.4) | |
| Brick | 91 (55.5) | 25 (15.2) | |
| Total family income, BDT, median (Q1;Q3) | 16,250 (12000; 25000) | 10,000 (8000; 13000) | <0.001‡ |
| Income group, n (%) | | | |
| Low income | 1 (0.6) | 14 (8.5) | <0.001† |
| Lower-middle income | 103 (62.8) | 141 (86.0) | |
| Upper-middle income | 55 (33.5) | 8 (4.9) | |
| High income | 5 (3) | 1 (0.6) | |

BDT = Bangladeshi Taka; 1US$ = 80 BDT

Results are expressed as number (%), mean (±SD) and median (Q1;Q3);

*Paired t-test was performed for paired continuous, normally distributed variables;

†McNemar $X^2$test was performed for paired categorical variables;

‡Wilcoxon signed-rank test was performed for paired continuous, non-normally distributed variables.

vs 33.4±9.3; p = 0.02). The percentage of those with university level education was significantly higher for rural-to-urban migrants than their rural siblings whereas nearly one-third of rural siblings (31.7%) were illiterate to primary level, compared to 14.6% of migrants (p < .001). About half of the migrants (47%) were professional workers, and the proportion of manual workers was lower for migrants than their rural counterparts (53% vs 65.9%; p<0.02). Migrants had higher average incomes compared to rural siblings. More than a third of migrants were classified in the upper middle income or high income category compared to 5% in the rural sibling group.

## CVD risk factors and migration status

For most of the risk factors, migrants had higher levels than the rural group except smokeless tobacco consumption. No significant difference was observed for waist: hip ratio, hypertension and TG between migrant and rural siblings (Fig 2).

Table 2 shows the β coefficients (95% CI) of continuous cardiovascular risk factors by migration status. Migrants watched television on average 20 minutes (95% CI 6.17–35.08 min/day) more per day than the rural sibling group whereas PUFA intake, fruit and vegetable and fish intake of the migrants were -5.31 gm/day (-6.91; -3.70), -21.64 serving/week (-28.20;

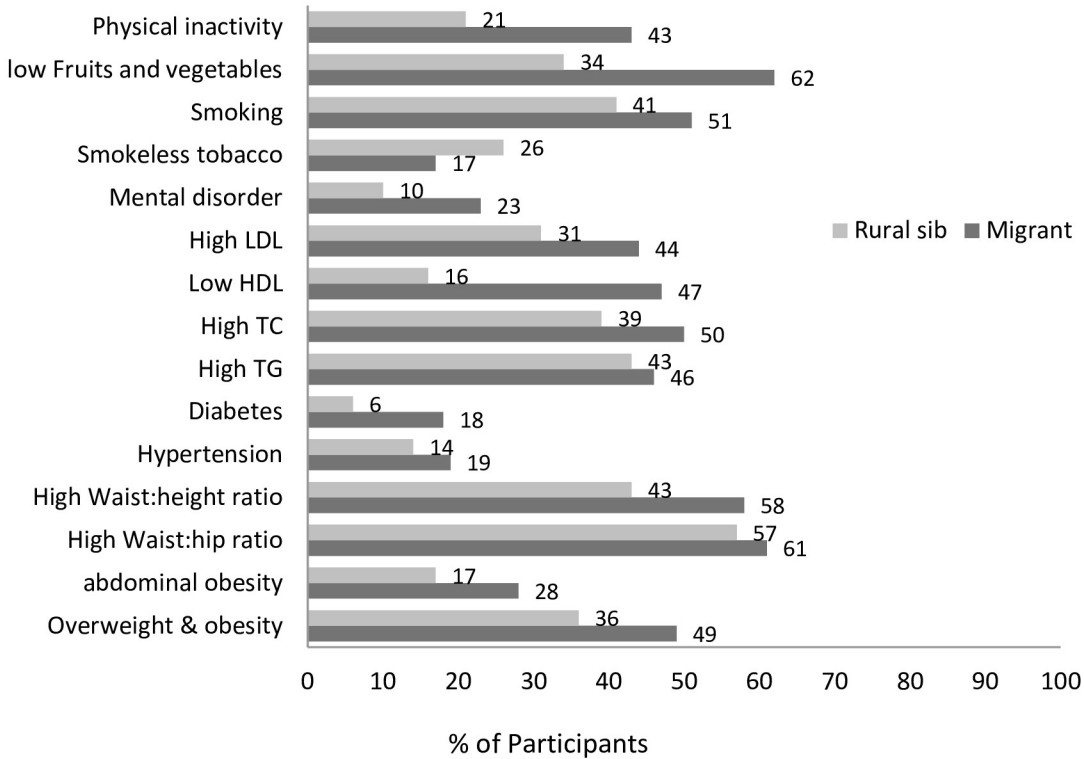

**Fig 2. Proportion of CVD risk factors by study groups.**

-15.09), -14.10 serving/week (-18.32; -9.87), respectively, lower than that of the rural siblings. With the exception of systolic and diastolic blood pressure, migrants had a consistently greater number of adverse measures than their rural siblings; this includes BMI, skinfold thickness, HDL, TC/HDL ratio, fasting blood glucose and 2-hr OGTT after adjusting for all confounders. These measures were categorised as presence or absence of each CVD risk factor and Table 3 presents the adjusted odds ratios for these risk factors among migrants compared to rural siblings. Physical inactivity, inadequate intake of fruit and vegetables and possible existence of a mental health disorder had 3.3 (1.73; 6.16), 4.3 (2.32; 7.92) and 2.9 (1.37; 6.27) times higher odds among migrants than their rural siblings. Furthermore, in the first multivariate model, migrants had 3.92 (95% CI: 2.03; 7.56) times higher odds to have low HDL levels and 3.69 (95% CI: 1.46; 9.35) times higher odds to be diabetic compared to their rural siblings. However, attenuation was observed for diabetes after adjustment for BMI, smoking, tobacco use, and mental disorders in model 3 and 4. Migrants had a 50% higher odds of being classified as hypertensive than their rural siblings (AOR = 1.5, 95% CI: 0.63–3.83). Although adjusted mean SBP and DBP were lower in migrants than in their rural siblings (Table 2), this pattern was reversed when hypertension was the outcome after adding the current treatment with antihypertensive drug (Table 3).

## Migration and acculturation

When participants were asked which dialect they used to communicate with their spouse, children, parents and friends, nearly all (96%) migrants communicated in the local dialect with their parents but this number decreased to one-third while communicating with their spouse, children and friends. Only 10% listened to the local music (their rural origin) and 90% liked

**Table 2. β coefficients (95% CI) of continuous cardiovascular risk factors by migration status.**

| Variables | Rural siblings | Rural-to-Urban Migrants |
|---|---|---|
| | | β coefficients (95% CI) |
| **MVPA MET-min/week** | | |
| Model 1 | Ref | -31.1 (-187.96; 6.50) |
| **TV watching, min/day** | | |
| Model 1 | Ref | 20.6 (6.17; 35.08) |
| **PUFA intake, gm/day** | | |
| Model 1 | Ref | -5.3 (-6.91; -3.70) |
| **Fruit & veg intake, serving/week** | | |
| Model 1 | Ref | -21.6 (-28.20; -15.09) |
| **Fish intake, serving/week** | | |
| Model 1 | Ref | -14.1 (-18.32; -9.87) |
| **BMI, kg/m$^2$** | | |
| Model 1 | Ref | 0.8 (0.02; 1.67) |
| Model 2 | Ref | 1.2 (0.37; 2.13) |
| **Waist circumference, cm** | | |
| Model 1 | Ref | 2.8 (0.02; 5.59) |
| Model 2 | Ref | 2.9 (-0.91; 5.92) |
| **All Skinfolds, mm** | | |
| Model 1 | Ref | 14.2 (9.13; 19.33) |
| Model 2 | Ref | 17.2 (11.49; 22.95) |
| **TG, mgm/dl** | | |
| Model 1 | Ref | 7.7 (-17.95; 33.29) |
| Model 2 | Ref | 12.5 (-16.47; 41.40) |
| Model 3 | Ref | 5 (-23.39; 33.47) |
| **TC, mgm/dl** | | |
| Model 1 | Ref | 11.4 (-2.34; 25.91) |
| Model 2 | Ref | 9.4 (-6.20; 25.06) |
| Model 3 | Ref | 4.2 (-10.87; 19.29) |
| **HDL, mgm/dl** | | |
| Model 1 | Ref | -2.8 (-4.79; -0.84) |
| Model 2 | Ref | -2.9 (-5.11; -0.66) |
| Model 3 | Ref | -2.7 (-4.92; -0.43) |
| **LDL, mgm/dl** | | |
| Model 1 | Ref | 16.4 (4.47; 28.38) |
| Model 2 | Ref | 13.9 (0.27; 27.48) |
| Model 3 | Ref | 9.9 (-3.36; 23.20) |
| **TC/HDL ratio** | | |
| Model 1 | Ref | 0.7 (0.33; 1.15) |
| Model 2 | Ref | 0.7 (0.25; 1.18) |
| Model 3 | Ref | 0.5 (0.11; 1.0) |
| **Fasting Blood Glucose (FBG), mmol/l** | | |
| Model 1 | Ref | 0.6 (0.27; 0.85) |
| Model 2 | Ref | 0.6 (0.25; 0.92) |
| Model 3 | Ref | 0.5 (0.16; 0.83) |
| Model 4 | Ref | 0.5 (0.21; 0.87) |
| **2-hr OGTT, mmol/l** | | |
| Model 1 | Ref | 1.2 (0.69; 1.63) |

(*Continued*)

**Table 2.** (Continued)

| Variables | Rural siblings | Rural-to-Urban Migrants |
|---|---|---|
| | | β coefficients (95% CI) |
| Model 2 | Ref | 1.1 (0.54; 1.61) |
| Model 3 | Ref | 0.9 (0.37; 1.43) |
| Model 4 | Ref | 0.9 (0.39; 1.46) |
| **Systolic, mmHg** | | |
| Model 1 | Ref | -4.7 (-7.89; -1.51) |
| Model 2 | Ref | -3.9 (-7.52; -0.47) |
| Model 3 | Ref | -5.4 (-8.75; -2.08) |
| Model 4 | Ref | -5 (-8.40; -1.63) |
| **Diastolic, mmHg** | | |
| Model 1 | Ref | -2.9 (-5.27; -0.45) |
| Model 2 | Ref | -1.9 (-4.65; 0.71) |
| Model 3 | Ref | -2.9 (-5.54; -0.42) |
| Model 4 | Ref | -2.8 (-5.40; 0.21) |

BMI = Body Mass Index, TG = Triglyceride, TC = Total Cholesterol, HDL = High-density lipoprotein, LDL = Low-density lipoprotein, OGTT = Oral Glucose Tolerance Test

Linear Mixed Effect Model was performed with a random effect of sibling-pair

Model 1: Adjusted for Age, Marital status, Education, Occupation, House type, Monthly family income

Model 2: as model 1 + Energy intake, MET-min/week

Model 3: as model 2 + BMI

Model 4: as model 3 + Smoking, Tobacco intake, Family history, and Mental disorder

other types of music. When migrants were asked if their dietary habits changed since migrating the city, most of them reported increasing consumption of unhealthy foods e.g., soft drinks (74%), energy drinks (59%), coffee/tea (60%), processed/canned foods (76%), eating out (72%) and red meat (60%). Vegetables were consumed less often compared to before migration (56%). Participants prepared their traditional food less often (19.5%) or not at all (28.7%), compared to before migration. Regarding perception of physical activity habit more than half (59%) of the participants indicated that they were more active compared to before immigration whereas 36% believed that they were less active (Table 4).

Table 5 shows exposure to urban life years associated with CVD risk factors, unadjusted and adjusted for a range of confounders. Urban life years were categorised into tertiles and a significant trend for higher levels of risk factors, except tobacco and hypertension, from rural non-migrants to migrants was seen for the unadjusted OR. After adjusting, the odds of physical inactivity, inadequate fruit and vegetable intake, a mental health disorder and low HDL were significantly higher in migrants than in rural siblings and for low HDL tended to be higher for each increasing tertile of urban life exposure. However, diabetes was 5.5 times higher in migrants than rural siblings in the first tertile of urban life exposure, but no clear pattern of higher diabetes risk was found in the following tertiles.

## Discussion

To the best of our knowledge this is the first study to look at the relationship between internal migration from rural to urban areas and CVD risk factors among the Bangladeshi population. In this study migration was found to be associated with an increase in physical inactivity and reduced fruit and vegetable and PUFA intake in migrants, compared with rural siblings, and

**Table 3. Adjusted odds ratios (95% CI) for cardiovascular risk factors by migration status.**

| Risk Factors | Rural siblings | Rural-to-Urban Migrants Odds ratios (95% CI) |
|---|---|---|
| **Physically inactive** | | |
| Model 1 | Ref | 3.3 (1.73; 6.16) |
| **Inadequate fruit & veg intake ($\leq$5 serving/ day)** | | |
| Model 1 | Ref | 4.3 (2.32; 7.92) |
| **Ever smoked** | | |
| Model 1 | Ref | 1.5 (0.84; 2.56) |
| **Smokeless tobacco intake** | | |
| Model 1 | Ref | 0.9 (0.40; 1.77) |
| **Mental health disorder[†]** | | |
| Model 1 | Ref | 2.9 (1.37; 6.27) |
| **Overweight and obesity** | | |
| Model 1 | Ref | 1.1 (0.62; 2.06) |
| Model 2 | Ref | 1.3 (0.66; 2.58) |
| **Abdominal obesity** | | |
| Model 1 | Ref | 1.2 (0.62; 2.42) |
| Model 2 | Ref | 1.4 (0.64; 2.99) |
| **High TC** | | |
| Model 1 | Ref | 1.2 (0.66; 2.15) |
| Model 2 | Ref | 1.3 (0.58; 2.22) |
| Model 3 | Ref | 0.9 (0.49; 1.98) |
| **Low HDL** | | |
| Model 1 | Ref | 3.9 (2.03; 7.56) |
| Model 2 | Ref | 3.6 (1.72; 7.66) |
| Model 3 | Ref | 3.5 (1.66; 7.49) |
| **High LDL** | | |
| Model 1 | Ref | 1.3 (0.68; 2.40) |
| Model 2 | Ref | 1.3 (0.65; 2.72) |
| Model 3 | Ref | 1.2 (0.57; 2.51) |
| **Diabetic[*†]** | | |
| Model 1 | Ref | 3.7 (1.46; 9.35) |
| Model 2 | Ref | 3.1 (1.06; 8.86) |
| Model 3 | Ref | 2.5 (0.89; 7.15) |
| Model 4 | Ref | 2.1 (0.711; 6.17) |
| **Hypertension[‡]** | | |
| Model 1 | Ref | 1.7 (0.83; 3.56) |
| Model 2 | Ref | 1.7 (0.73; 3.84) |
| Model 3 | Ref | 1.5 (0.63; 3.57) |
| Model 4 | Ref | 1.5 (0.63; 3.83) |

[*] Diabetes defined if FBS $\geq$7.0 mmol/L or OGTT $\geq$11.1 mmol/L or self-reported diabetes medication use;

[‡]Hypertension defined if systolic blood pressure $\geq$ 140 mmHg or a diastolic blood pressure $\geq$ 90 mmHg or current treatment with antihypertensive medication

Generalized Linear Model was performed including a random effect of sibling pairs;

[†]excluding random effect as no variation between sibling pair

Model 1: Adjusted for Age, Marital status, Education, Occupation, House type, Monthly family income

Model 2: as model 1 + Energy intake, Physical activity level

Model 3: as model 2 + BMI

Model 4: as model 3 + Smoking, Tobacco intake, and Mental disorder

**Table 4. Differences in level of acculturation among rural-to-urban migrants.**

| Acculturation variables | Rural-to-Urban Migrants | | | |
|---|---|---|---|---|
| **Use of Language, n (%)** | **Local dialect** | **Both equally** | **Standard dialect** | **N/A** |
| **Communicates with** | | | | |
| Spouse | 60 (36.6) | 18 (11.0) | 39 (23.8) | 47 (28.7) |
| Children | 44 (26.9) | 15 (9.1) | 34 (23.1) | 67 (40.9) |
| Parents | 149 (90.9) | 3 (1.8) | 4 (2.4) | 8 (4.9) |
| Friends | 43 (26.2) | 60 (36.6) | 61 (37.2) | 0 (0) |
| **Favourite music** | | | | |
| Local music | 16 (9.8) | | | |
| Other | 148 (90.2) | | | |
| **Dietary Practice, n (%)** | **More often** | **Same** | **Less often** | **Not at all** |
| **Current Dietary habit compared to before migration** | | | | |
| Soda/ soft drinks | 121 (73.8) | 22 (13.4) | 16 (9.8) | 5 (3.0) |
| Energy drinks | 97 (59.1) | 13 (7.9) | 8 (4.9) | 46 (28.0) |
| Coffee/tea | 98 (59.8) | 24 (14.6) | 29 (17.7) | 13 (7.9) |
| Plain water | 98 (59.8) | 42 (25.6) | 24 (14.6) | 0 (0) |
| Fast food | 69 (42.1) | 4 (2.4) | 2 (1.2) | 89 (54.3) |
| Oily local foods | 93 (56.7) | 19 (11.6) | 41 (25.0) | 11 (6.7) |
| Chips/popcorn | 78 (47.6) | 13 (7.9) | 32 (19.5) | 41 (25.0) |
| Processed or canned foods | 124 (75.6) | 3 (1.8) | 2 (1.2) | 35 (21.3) |
| Eating out | 118 (72.0) | 5 (3.0) | 16 (9.8) | 25 (15.2) |
| Butter/cheese/ mayonnaise/ ghee | 26 (15.9) | 3 (1.8) | 33 (20.1) | 102 (62.2) |
| Fruits | 92 (56.1) | 8 (4.9) | 64 (39.0) | 0 (0) |
| Vegetables | 61 (37.2) | 12 (7.3) | 91 (55.5) | 0 (0) |
| Beef/mutton | 99 (60.4) | 13 (7.9) | 50 (30.5) | 2 (1.2) |
| Chicken | 134 (81.7) | 7 (4.3) | 22 (13.4) | 1 (0.6) |
| Fish | 115 (70.1) | 14 (8.5) | 35 (21.3) | 0 (0) |
| Taking vitamin supplement | 65 (39.6) | 5 (3.0) | 10 (6.1) | 84 (51.2) |
| **Preparation of typical dishes of origin** | 48 (29.3) | 37 (22.6) | 32 (19.5) | 47 (28.7) |
| **Perception of Physical Activity, n (%)** | **much more active** | **more active** | **same** | **less active** | **much less active** |
| **Current physical activity habit compared to before migration** | 41 (25.0) | 57 (34.8) | 7 (4.3) | 47 (28.7) | 12 (7.3) |

Results are expressed as number (%)

this likely contributed to the higher levels of BMI, skinfold thickness and lower HDL in migrants. A trend of higher levels of risk in migrants compared to rural non-migrants was seen with a longer period of stay in the urban area for physical inactivity, inadequate fruit and vegetable intake, a mental health disorder and low HDL. Separate analyses selecting only migrants showed that for the majority, language, dietary habits and physical activity have changed after migration due to acculturation.

## CVD risk factors by migration status

This study confirms that low levels of physical activity and high sedentarism were more prevalent amongst urban migrants compared to their rural siblings, which supports the causal effect of migration rather than selection by pre-migration risk. The findings are consistent with migrant studies in India, Peru, Guatemala and Tanzania [13, 15, 43, 44]. Very few migration studies [15, 44] have explored sedentary behaviour among migrants. In this study, duration of

**Table 5. CVDs risk factors by tertiles of urban life-years.**

| Risk Factors | 0 urban life years | 1–6 urban life years | 7–12 urban life years | >12 urban life years | p for Trend |
|---|---|---|---|---|---|
| Unadjusted OR (95% CI) | | | | | |
| Physical inactivity | ref | 2.4 (1.24; 4.48) | 2.6 (1.32; 5.05) | 3.4 (1.78; 6.57) | <0.001 |
| Inadequate fruit & veg intake (≤5 serving/day) | ref | 5.2 (2.68; 10.01) | 2.2 (1.15; 4.11) | 2.6 (1.39; 4.87) | 0.001 |
| Ever smoked | ref | 1.3 (0.72; 2.38) | 1 (0.54; 1.92) | 2.7 (1.41; 5.06) | 0.012 |
| Smokeless tobacco intake | ref | 0.4 (0.19; 1.01) | 0.7 (0.32; 1.49) | 0.6 (0.30; 1.38) | 0.17 |
| Mental health disorder | ref | 2.2 (0.98; 4.96) | 3.9 (1.82; 8.59) | 1.7 (0.72; 4.15) | 0.02 |
| Overweight and obesity | ref | 0.9 (0.53; 1.83) | 1.7 (0.91; 3.23) | 3.3 (1.72; 6.24) | <0.001 |
| Abdominal obesity | ref | 0.9 (0.45; 2.19) | 1 (0.46; 2.38) | 4.9 (2.48; 9.50) | <0.001 |
| Low HDL | ref | 3.3 (1.60; 6.58) | 4.4 (2.14; 9.09) | 7.6 (3.74; 15.52) | <0.001 |
| Diabetes | ref | 3.5 (1.38; 9.07) | 1.7 (0.53; 5.28) | 4.2 (1.66; 10.72) | 0.007 |
| Hypertension | ref | 1.7 (0.81; 3.69) | 0.9 (0.39; 2.43) | 1.6 (0.71; 3.48) | 0.376 |
| Adjusted OR (95% CI) | | | | | |
| Physical inactivity | ref | 2.9 (1.34; 6.06) | 3.6 (1.62; 7.95) | 3.6 (1.57; 8.39) | 0.001 |
| Inadequate fruit & veg intake (≤5 serving/day) | ref | 5.5 (2.64; 11.63) | 2.7 (1.30; 5.69) | 4.8 (2.13; 10.98) | <0.001 |
| Ever smoked | ref | 1.7 (0.85; 3.43) | 0.9 (0.46; 1.99) | 1.9 (0.87; 4.16) | 0.22 |
| Smokeless tobacco intake | ref | 1 (0.39; 2.79) | 0.9 (0.34; 2.15) | 0.6 (0.23; 1.68) | 0.37 |
| Mental health disorder | ref | 2.1 (0.86; 5.27) | 4.8 (1.93; 11.88) | 2.6 (0.92; 7.60) | 0.009 |
| Overweight and obesity | ref | 1 (0.50; 2.10) | 1.1 (0.51; 2.32) | 1.2 (0.54; 2.74) | 0.64 |
| Abdominal obesity | ref | 1.1 (0.44; 2.58) | 0.7 (0.28; 1.83) | 2.2 (0.93; 5.07) | 0.16 |
| Low HDL | ref | 2.7 (1.23; 6.02) | 4.1 (1.80; 9.16) | 6.5 (2.74; 15.59) | <0.001 |
| Diabetes | ref | 5.5 (1.81; 16.88) | 2.1 (0.57; 7.71) | 3.3 (0.97; 11.17) | 0.12 |
| Hypertension | ref | 2.3 (0.95; 5.40) | 1.2 (0.42; 3.18) | 1.6 (0.61; 4.44) | 0.38 |

*Non-migrants are in the zero urban life years group. Rest of the groups are categorized in tertiles

Adjusted for Age, Marital status, Education, Occupation, House type, Monthly family income

sitting was significantly higher in migrants than their rural siblings, and TV watching was an estimated average 20 minutes/day higher in migrants compared to their rural siblings. This is in line with the Indian Migration Study (IMS) where migrant men reported one hour more sedentary behaviour and 30 minutes more television viewing than their rural siblings (per day) [44].

The results from this study indicate that migration from a rural area to a large city such as Dhaka, can have an impact on migrants' diet, as also documented by other studies [12, 13, 43, 45–47]. This analysis found that migration from rural to urban areas was associated with lower PUFA intake and less frequent consumption of fruit and vegetables and fish. The most probable reason for less frequent consumption are the higher prices of fruit and vegetables in urban areas [20, 48]. In rural areas, people meet their dietary needs by cultivating fruit and seasonal vegetables in their home gardens. However, most rural-to-urban migrants live in rented houses (83% in this study) and there might be no opportunity or time to do gardening. Moreover, local varieties of seasonal fruits are sometimes not considered as good fruit by urban people whereas imported, costly fruits are considered real fruit but these are beyond their budget [20].

The effect of less frequent consumption of fruits and vegetables might be observed the body fat and lipid level of participants. Although the magnitude of the difference in BMI was not that high (β = 1.24 kg/m²) between groups, skinfold thickness were significantly different (β = 17.22mm) between the study groups. Similar findings have been reported in studies from Peru [49] and Guatemala [13]. These particular characteristics, (i.e, low BMI and excess body fat),

have also observed in other studies of Indian Asians [50, 51]. Besides this genetic trait, food habits and marked decrease in physical activity might be possible important reasons for the significant difference in body fat among migrant and rural siblings. Migrants also had a higher prevalence of dyslipidemia than their rural counterparts. Migrants were around 3 times higher odds to have low HDL levels than their rural siblings. It seems possible that these results are due to the low intake of fruit, vegetables and fish, which leads to low PUFA intake and subsequently worsened serum lipid profile in migrants.

For blood pressure, while the mean value of systolic and diastolic BP were higher in rural siblings than rural-to-urban migrants, this trend was reversed after including treatment with antihypertensive drug. It indicates that more migrants might be on treatment to manage their blood pressure and, thus, pushing the mean blood pressure of migrants to be lower compared to their rural siblings. However, a systematic review on this topic also indicated the inconsistent pattern of hypertension in the previous migration studies [52].

In this study, migrants were found to have three times higher odds of developing a mental health disorder than their rural siblings after adjusting for demographic variables. Mental illness may be due to an underlying genetic disposition, however, here we partially control for this by looking at dependant pairs, suggesting environmental factors influencing migrants may have driven the differences between siblings. After moving from a rural area to urban life, adjustment and settlement in the urban area may induce stress. Other factors could be missing family and social networks, reduced social support or economic deprivation [53]. Another rural-to-urban cross-sectional migration study conducted in Dhaka and adjacent rural area of Bangladesh also reported that the prevalence of poor mental condition was higher in migrants (60%) than rural residents (39%) and the urban group (54%) [53].

## Migration, acculturation and CVD risk factors

Language spoken at home is considered as a proxy measure of acculturation with standard Bangla the language of Dhaka and local dialects used in Pirganj. This study suggests acculturation was under way in migrants as around two-third of migrants used only standard dialect or both standard or local dialect of Bangla language to communicate with their spouse or children and friends. Although most of the migrants (90%) used local dialect with their parents, it decreased to one-third while communicating with their spouse, children and friends. In case of diet, dietary changes following migration emerge as a significant indication of acculturation.

Dietary changes may occur in many forms such as choosing fast food or processed food over traditional food for convenience, and changes to cooking habits especially using the traditional recipe or changes to meal formats [54]. In this study, more the majority stated that their diet became unhealthy and abandoned traditional food. Although we have asked only one question regarding physical activity which cannot reflect the context of physical activity change (i.e., leisure or work, intensity and frequency), more than one-third of rural-to-urban migrants believed they were less active than before migration.

Length of urban residence is widely used as a proxy measure of acculturation. In this study a general trend of higher levels of risk from rural non-migrants to urban migrants was observed. The risk of physical inactivity, inadequate fruit and vegetable intake, mental health disorder and low HDL were significantly higher in migrants than in rural siblings, with a tendency for physical inactivity and low HDL to be higher with increased exposure to urban life. Diabetes and hypertension did not show any significant gradient in the adjusted model, perhaps due to the small number of diagnosed cases. No consistent pattern was observed for hypertension in the published literature, with some studies showing length of residence in urban areas was positively associated with hypertension [46, 55, 56], while other studies

reported recent rural-to-urban migrants were more likely to be hypertensive than long-term migrants [57, 58]. The latter pattern may be related to the stress associated with migration, which eventually resolves by treatment or by developing resiliency. In the Indian Migration Study (IMS), although a trend for higher levels of risk factors from non-migrants to migrants was observed for TC, TG and diabetes, no clear pattern was observed for HDL [59]. Other studies [46, 55, 58, 60, 61] usually focused on the metabolic risk factors for CVD and there is a scarcity of studies on unhealthy diet, physical activity and acculturation.

## Strength and limitations

Longitudinal design is seldom feasible and therefore most internal migration studies use cross-sectional comparison [52]. However, the strength of the sibling-pair comparative design used here is to examine the impact of migration within siblings that share similarities in origin including genetics, rearing environment, culture, exposures to diet, micro-and macro rural environment. This allows for attribution of changes in the rural-to-urban migrants to the new environment interacting with new personal behavioural choices. An additional strength is the coverage of all established CVD risk factors in the same sample as well as using validated tools. Nevertheless, there are some shortcomings to this design. First, this study had lower response rates (63%) than we anticipated largely because of the complexity of the sibling-pair recruitment. This sibling-pair design is not familiar in Bangladesh and we need to get consent from both siblings to recruit them to the study. Another limitation is the lack of representation of women migrants. Therefore, our finding is only generalised to rural-to-urban male migrants. Initially we planned to include both genders. However, in rural Bangladesh, women usually move to their husband's home after marriage which is sometimes far from their parents' home or other town. Thus, when we tried to recruit both siblings, we could not reach rural female siblings. Moreover, family members were unwilling to give contact details of female migrants because of safety issues, which was not the case for male migrants. This is a methodological issue of recruitment and should be considered in planning further research on rural-to-urban female migrants studies. We could have designed a stratified sampling procedure based on quota by gender. However, this would complicate the recruitment even further given migration to city is largely dominated by men and often women followed.

## Conclusion

The findings suggest that men who migrated from rural-to-urban environment are at increased CVD risk which exacerbate with time spent in urban area due to acculturation. This study gives new insights into the risks related with migration and urbanization in Bangladesh. This will assist planning effective CVD preventive interventions. Such intervention can start before migration and continue after migration with mHealth-based interventions by assessing individual CVD risk, providing educational content and individualized reinforcement strategies, monitoring and feedback to promote long-term self-management. Moreover, as Bangladesh is the world's leading clothing exporter and most garment workers are rural migrants [62], special attention should be paid to promote healthy lifestyle within this industry. A study on rural-to urban women migrants is also warranted because their risk of CVD is different to men and their process of adaptation and acculturation to urban life may be different to men.

## Supporting information

**S1 File. Sampling technique.**
(DOCX)

## Acknowledgments

The authors wish to thank all participants of the study, data collectors and research assistants. Our appreciation also goes to Farhana Afroz Chowdhury Lopa, Ayesha Akter for their contribution to the study. Last but not least, we would like to acknowledge Tanvir Ahmed for his continuous support during the study period.

## Author Contributions

**Conceptualization:** Shirin Jahan Mumu, A. K. M. Fazlur Rahman, Paul P. Fahey, Liaquat Ali, Dafna Merom.

**Data curation:** Shirin Jahan Mumu, Dafna Merom.

**Formal analysis:** Shirin Jahan Mumu, Paul P. Fahey, Dafna Merom.

**Funding acquisition:** Shirin Jahan Mumu, A. K. M. Fazlur Rahman, Liaquat Ali, Dafna Merom.

**Investigation:** Shirin Jahan Mumu, A. K. M. Fazlur Rahman, Liaquat Ali, Dafna Merom.

**Methodology:** Shirin Jahan Mumu, Paul P. Fahey, Liaquat Ali, Dafna Merom.

**Project administration:** Shirin Jahan Mumu, A. K. M. Fazlur Rahman, Liaquat Ali, Dafna Merom.

**Resources:** Shirin Jahan Mumu, A. K. M. Fazlur Rahman, Liaquat Ali.

**Software:** Shirin Jahan Mumu, Dafna Merom.

**Supervision:** Shirin Jahan Mumu, A. K. M. Fazlur Rahman, Liaquat Ali, Dafna Merom.

**Validation:** Shirin Jahan Mumu, Dafna Merom.

**Visualization:** Shirin Jahan Mumu, Paul P. Fahey, Dafna Merom.

**Writing – original draft:** Shirin Jahan Mumu, Dafna Merom.

**Writing – review & editing:** Shirin Jahan Mumu, A. K. M. Fazlur Rahman, Paul P. Fahey, Dafna Merom.

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
