## [Decision Letter · Decision Letter 0]

20 Apr 2022

PONE-D-22-06449Lifestyle risk factors and metabolic markers of cardiovascular diseases in Bangladeshi rural-to-urban male migrants compared with their non-migrant siblings: a sibling-pair comparative studyPLOS ONE

Dear Dr. Mumu,

Thank you for submitting your manuscript to PLOS ONE. After careful consideration, we feel that it has merit but does not fully meet PLOS ONE’s publication criteria as it currently stands. Therefore, we invite you to submit a revised version of the manuscript that addresses the points raised during the review process.

We look forward to receiving your revised manuscript.

Kind regards,

Samantha Frances Ehrlich

Academic Editor

PLOS ONE

Journal Requirements:

3. PLOS requires an ORCID iD for the corresponding author in Editorial Manager on papers submitted after December 6th, 2016. Please ensure that you have an ORCID iD and that it is validated in Editorial Manager. To do this, go to ‘Update my Information’ (in the upper left-hand corner of the main menu), and click on the Fetch/Validate link next to the ORCID field. This will take you to the ORCID site and allow you to create a new iD or authenticate a pre-existing iD in Editorial Manager. Please see the following video for instructions on linking an ORCID iD to your Editorial Manager account: https://www.youtube.com/watch?v=_xcclfuvtxQ.

Reviewers' comments:

Reviewer's Responses to Questions

**Comments to the Author**

1. Is the manuscript technically sound, and do the data support the conclusions?

Reviewer #1: Yes

Reviewer #2: Yes

2. Has the statistical analysis been performed appropriately and rigorously? 

Reviewer #1: Yes

Reviewer #2: Yes

3. Have the authors made all data underlying the findings in their manuscript fully available?

Reviewer #1: Yes

Reviewer #2: Yes

4. Is the manuscript presented in an intelligible fashion and written in standard English?

Reviewer #1: Yes

Reviewer #2: Yes

5. Review Comments to the Author

Reviewer #1: The reviewed manuscript is based on original data, very actual and includes results of very interesting research topic. As indicates the title of the manuscript, the results of the manuscript are based only on the examination data of male sibling-pairs. Therefore I recommend indicate in the aim of the manuscript that only male sibling-pair data were analyzed and exclude from the Method section text and other parts of the manuscript, in which female sibling-pair selection, numbers of female sibling-pairs the exclusion criteria (for example, pregnant women) (line 153) and so on, were mentioned. You mention in the text of the manuscript term "CVDs risk factors": physical inactivity, smoking, overweight and obesity, abdominal obesity, low HDL. I did not find in the section Methods described how those factors were determined and classified. The only definition of hypertension and diabetes are presented (lines 329-330). So, please include the description of the methods of the examination and classification of the CVDs risk factors into the Methods section. Maybe you presented the mentioned description of the determination and classification of the CVDs risk factors in your previous publications: so, you could indicate such publication in the list of the references instead of including of additional paragraph into the section Methods of the manuscript.

Reviewer #2: This is an interesting work on male (in-country) rural-to-urban migrants on a major public health issue in Bangladesh, NCD risk factors. This study has shown how exposure of the people to an urban environment (physical and others) gradually develops acquire risk factors to develop CVD in future compared to their rural sibs. This is probably done for the first time in Bangladesh. The authors deserve special felicitation for this milestone piece.

The manuscript is well written and well presented. However, it needs some corrections and revisions for further improvements.

Comments:

1. Line 94: Four STEPS surveys have been mentioned, but two have been cited. Reference to 2006 (Indian Heart Journal) and 2010 (Indian Journal of Public Health) could be made.

2. Line 110: SES has appeared for the first time without giving the full name.

3. Line 118: Share lifestyle has been mentioned, but the issue goes beyond. The environment could be considered.

4. Lines 138-146: Provides 2011 Census data on the population size. Having a projected population close to the survey time could be more informative. I believe BBS has such data, online or otherwise.

5. Line 185: Reference has been made to the Supplementary file about the sampling technique. It could be mentioned briefly here also so that readers can understand it without reading the Supplementary file.

6. Lines 187-242: Variable ascertainment has been given in these lines.

a. PUFA has been mentioned in the Results, Discussion, and Abstract. Therefore, PUFA estimation should be mentioned here.

b. Moreover, the family income data has been presented. How was it determined for the rural sibs? Estimation of income in a rural area is complex and unclear if multiple questions are not included. How valid is the authors’ data on income? How did you classify the economic classes of the study participants: low, lower-middle, upper-middle, and high income?

7. Line 142: Friedwald formula could have a citation here.

8. Line 260: “Data presented in tables and graphs” could be dropped.

9. Line 276: Results up to one decimal could suffice without losing any information.

10. Table 1: Treatment history for diabetes and hypertension could be one of the determinants of their blood glucose and blood pressure levels. This has been mentioned by the authors in the Discussion section. Can we have the treatment history data here?

11. Table 2: Given the authors have emphasized their findings on skinfold thickness and PUFA but presented the results in the Supplementary file, that does not come to the readers’ notice immediately.

a. Therefore, PUFA and skinfold thickness should be presented here as has been done for truncated quantitative variables.

b. Lines 318-320: Attenuation of ORs, in fact, became significant for models 3 and 4. These two models include BMI, smoking, tobacco use, and mental disorders. Please revise the stamen.

12. Table 4: It has been inadvertently labeled as Table 1. All ORs and their confidence intervals could be presented up to one decimal point. This will not lose any information, but the Table will appear clear and succinct. Please consider this for Table 2 also.

13. Lines 434 & 436: The official language of Bangladesh is Bangla, not Bengali. Consider revising it.

14. Lines 452: Clear gradient was not observed for adjusted ORs for fruit/veg intake, smoking, and mental disorders in addition to diabetes and hypertension (Table 4). Please revise.

15. Lines 485-487: This statement better suits the Conclusion. Please take it there.

16. References: Kindly use URL for the websites and DOI for journal articles (as much as available), unless it is not a requirement of the journal’s style.

17. Abstract: Please see the comments above about the PUFA, and decimal points.

18. Additional suggestion: Clustering of CVD risk factors is a feature in Bangladeshi people. It would be excellent having such data (>=3 risk factors as an example) in this article.

6. PLOS authors have the option to publish the peer review history of their article (what does this mean?). If published, this will include your full peer review and any attached files.

Reviewer #1: No

Reviewer #2: **Yes: **Professor M Mostafa Zaman, Ekhlaspur Center of Health, Chandpur, Bangladesh

---

## [Author Response · Author response to Decision Letter 0]

22 Jul 2022

Reviewer 1 comments

1 The reviewed manuscript is based on original data, very actual and includes results of very interesting research topic. As indicates the title of the manuscript, the results of the manuscript are based only on the examination data of male sibling-pairs. Therefore I recommend indicate in the aim of the manuscript that only male sibling-pair data were analyzed and exclude from the Method section text and other parts of the manuscript, in which female sibling-pair selection, numbers of female sibling-pairs the exclusion criteria (for example, pregnant women) (line 153) and so on, were mentioned. Included ‘male sibling-pair’ in the aim.

Line 153 is modified, and ‘pregnant women’ is deleted

Female were deleted from method. Figure 1 is modified as well 

 P 6 , 7 ; line 121, 152-153

Figure 1

2 You mention in the text of the manuscript term "CVDs risk factors": physical inactivity, smoking, overweight and obesity, abdominal obesity, low HDL. I did not find in the section Methods described how those factors were determined and classified. The only definition of hypertension and diabetes are presented (lines 329-330). So, please include the description of the methods of the examination and classification of the CVDs risk factors into the Methods section. Maybe you presented the mentioned description of the determination and classification of the CVDs risk factors in your previous publications: so, you could indicate such publication in the list of the references instead of including of additional paragraph into the section Methods of the manuscript. The classification of ‘CVD risk factors’ are included in the method section. P 12 ; line 253-268

Reviewer 2 comments

1 This is an interesting work on male (in-country) rural-to-urban migrants on a major public health issue in Bangladesh, NCD risk factors. This study has shown how exposure of the people to an urban environment (physical and others) gradually develops acquire risk factors to develop CVD in future compared to their rural sibs. This is probably done for the first time in Bangladesh. The authors deserve special felicitation for this milestone piece. We thank the reviewer for appreciating our work and indeed it is the first of its kind in Bangladesh. 

2 Line 94: Four STEPS surveys have been mentioned, but two have been cited. Reference to 2006 (Indian Heart Journal) and 2010 (Indian Journal of Public Health) could be made. Two have been cited because all 2006, 2010 and 2013 surveys are discussed in Zaman et al 2015 article. However, we have included 2006 and 2010 references as well. P 5 ; line 94-97

3 Line 110: SES has appeared for the first time without giving the full name. Thanks for alerting us, we now specified socio-economic status (SES) in the text P 5 ; line 110

4 Line 118: Share lifestyle has been mentioned, but the issue goes beyond. The environment could be considered. The line is modified as per reviewer’s suggestion 

‘Hence the differences observed at present between the migrant and the rural sibling represent divergence from the shared lifestyle and environment change’. 

 P 6 ; line 117-118

5 Lines 138-146: Provides 2011 Census data on the population size. Having a projected population close to the survey time could be more informative. I believe BBS has such data, online or otherwise. We have included last updated data from BBS website (15 January 2019; http://www.bbs.gov.bd/site/page/2888a55d-d686-4736-bad0-54b70462afda/-). 

P 7 ; line 144

6 Line 185: Reference has been made to the Supplementary file about the sampling technique. It could be mentioned briefly here also so that readers can understand it without reading the Supplementary file. This paragraph is modified as per reviewer’s suggestion P 9 ; line 181-186

7 Lines 187-242: Variable ascertainment has been given in these lines.

a. PUFA has been mentioned in the Results, Discussion, and Abstract. Therefore, PUFA estimation should be mentioned here.

b. Moreover, the family income data has been presented. How was it determined for the rural sibs? Estimation of income in a rural area is complex and unclear if multiple questions are not included. How valid is the authors’ data on income? How did you classify the economic classes of the study participants: low, lower-middle, upper-middle, and high income? We have now included PUFA estimation in the text.

We totally agree with the reviewer that it is critical to estimate income in a rural area. Indeed, we have asked multiple questions including monthly income and expenditure, yearly income and sources of income, such as farming, business, rent, remittance etc, and ownership of home, land and assets. 

Reference is included for the classification of economic classes.

 P 9, 12 ; line 189-194, 195-198, 253-254

8 Line 142: Friedwald formula could have a citation here. Reference is included in the text P 11 ; line 246-247

9 Line 260: “Data presented in tables and graphs” could be dropped. This line is removed. 

10 Line 276: Results up to one decimal could suffice without losing any information. We now changed the result of the estimates to one decimal place. P 14 ; line 296

11 Table 1: Treatment history for diabetes and hypertension could be one of the determinants of their blood glucose and blood pressure levels. This has been mentioned by the authors in the Discussion section. Can we have the treatment history data here? Treatment history data is now included in the analysis section P 12 ; line 266, 268

12 Table 2: Given the authors have emphasized their findings on skinfold thickness and PUFA but presented the results in the Supplementary file, that does not come to the readers’ notice immediately.

a. Therefore, PUFA and skinfold thickness should be presented here as has been done for truncated quantitative variables.

b. Lines 318-320: Attenuation of ORs, in fact, became significant for models 3 and 4. These two models include BMI, smoking, tobacco use, and mental disorders. Please revise the statement. a. Supplementary table and description are included in the result section as per reviewer’s suggestion 

b. The statement is revised. P 16, 17 ; line 335-342, 356, 350-351

13 Table 4: It has been inadvertently labeled as Table 1. All ORs and their confidence intervals could be presented up to one decimal point. This will not lose any information, but the Table will appear clear and succinct. Please consider this for Table 2 also. Table 4 label is corrected. Now it is Table 5

We now changed the result of the estimates to one decimal place and kept two decimal points in the 95% confidence intervals

 P 22 ; line 420

Table 2, 3, 5

14 Lines 434 & 436: The official language of Bangladesh is Bangla, not Bengali. Consider revising it. Revised and changed in the text P 9, 25 ; line 191, 485, 487

15 Lines 452: Clear gradient was not observed for adjusted ORs for fruit/veg intake, smoking, and mental disorders in addition to diabetes and hypertension (Table 4). Please revise. The line is revised.

‘The risk of physical inactivity, inadequate fruit and vegetable intake, mental health disorder and low HDL were significantly higher in migrants than in rural siblings, with a tendency for physical inactivity and low HDL to be higher with increased exposure to urban life’. P 26 ; line 501-503

16 Lines 485-487: This statement better suits the Conclusion. Please take it there. This line has now been moved to the conclusion P 28 ; line 549-551

17 References: Kindly use URL for the websites and DOI for journal articles (as much as available), unless it is not a requirement of the journal’s style. References are included following journal guideline 

18 Abstract: Please see the comments above about the PUFA, and decimal points. We now changed the result of the estimates to one decimal place. P 2 ; line 32-33

19 Additional suggestion: Clustering of CVD risk factors is a feature in Bangladeshi people. It would be excellent having such data (>=3 risk factors as an example) in this article. This is a good suggestion. However, this manuscript has already contained a lot of information as is. We will consider this analysis in another publication that is planned.

Further Table 4 is referred in the text

---

## [Decision Letter · Decision Letter 1]

26 Aug 2022

Lifestyle risk factors and metabolic markers of cardiovascular diseases in Bangladeshi rural-to-urban male migrants compared with their non-migrant siblings: a sibling-pair comparative study

PONE-D-22-06449R1

Dear Dr. Mumu,

We’re pleased to inform you that your manuscript has been judged scientifically suitable for publication and will be formally accepted for publication once it meets all outstanding technical requirements.

Kind regards,

Samantha Frances Ehrlich

Academic Editor

PLOS ONE

Additional Editor Comments (optional):

I wish to thank the authors for their careful consideration of the reviewers' comments, and to congratulate them on this work.

Reviewers' comments:

Reviewer's Responses to Questions

**Comments to the Author**

1. If the authors have adequately addressed your comments raised in a previous round of review and you feel that this manuscript is now acceptable for publication, you may indicate that here to bypass the “Comments to the Author” section, enter your conflict of interest statement in the “Confidential to Editor” section, and submit your "Accept" recommendation.

Reviewer #1: All comments have been addressed

Reviewer #2: All comments have been addressed

2. Is the manuscript technically sound, and do the data support the conclusions?

Reviewer #1: Yes

Reviewer #2: Yes

3. Has the statistical analysis been performed appropriately and rigorously? 

Reviewer #1: Yes

Reviewer #2: Yes

4. Have the authors made all data underlying the findings in their manuscript fully available?

Reviewer #1: Yes

Reviewer #2: Yes

5. Is the manuscript presented in an intelligible fashion and written in standard English?

Reviewer #1: Yes

Reviewer #2: Yes

6. Review Comments to the Author

Reviewer #1: All my questions were answered by authors of the manuscript. Authors of the manuscript modified the text of the manuscript according to all my recommendations. I recommend accept the manuscript.

Reviewer #2: The authors have addressed all poiys except the final point on clustering. Authors want to report it another article. I still suggest including a brief statement about the clustering. A graph could also be used to depict this. I believe, clustering has strong relationship with rural to urban migration. Therefore, it is very much needed here.

7. PLOS authors have the option to publish the peer review history of their article (what does this mean?). If published, this will include your full peer review and any attached files.

Reviewer #1: No

Reviewer #2: **Yes: **Professor M Mostafa Zaman, Ekhlaspur Center of Health, Matlab North, Chandpur, Bangladesh

---

## [Editor Report · Acceptance letter]

6 Sep 2022

PONE-D-22-06449R1 

Lifestyle risk factors and metabolic markers of cardiovascular diseases in Bangladeshi rural-to-urban male migrants compared with their non-migrant siblings: a sibling-pair comparative study 

Dear Dr. Mumu:

I'm pleased to inform you that your manuscript has been deemed suitable for publication in PLOS ONE. Congratulations! Your manuscript is now with our production department. 

Kind regards, 

on behalf of

Dr. Samantha Frances Ehrlich 

Academic Editor

PLOS ONE